# *N*,*N*-Diethyl-3-toluamide Formulation Based on Ethanol Containing 0.1% 2-Hydroxypropyl-β-cyclodextrin Attenuates the Drug’s Skin Penetration and Prolongs the Repellent Effect without Stickiness

**DOI:** 10.3390/molecules27103174

**Published:** 2022-05-16

**Authors:** Noriaki Nagai, Mayu Kawaguchi, Misa Minami, Kana Matsumoto, Tatsuji Sasabe, Kenji Nobuhara, Akira Matsubara

**Affiliations:** 1Faculty of Pharmacy, Kindai University, 3-4-1 Kowakae, Higashi-Osaka 577-8502, Japan; matsumoto-kana@kindai.ac.jp; 2Earth Corporation, 3218-12 Sakoshi, Ako 678-0192, Japan; kawaguchi-mayu@earth.jp (M.K.); mmm.1014.333@gmail.com (M.M.); sasabe-tatsuji@earth.jp (T.S.); nobuhara-kenji@earth.jp (K.N.); matsubara-akira@earth.jp (A.M.)

**Keywords:** *N*,*N*-diethyl-3-toluamide, cyclodextrin, insect repellent, stickiness, skin penetration

## Abstract

*N*,*N*-diethyl-3-toluamide (DEET) is one of the most widely used insect repellents in the world. It was reported that a solution containing 6–30% cyclodextrin (CD) as a solvent instead of ethanol (EtOH) provided an enhancement of the repellent action time duration of the DEET formulation, although the high-dose CD caused stickiness. In order to overcome this shortcoming, we attempted to prepare a 10% DEET formulation using EtOH containing low-dose CDs (β-CD, 2-hydroxypropyl-β-CD (HPβCD), methyl-β-CD, and sulfobutylether-β-CD) as solvents (DEET/EtOH/CD formulations). We determined the CD concentration to be 0.1% in the DEET/EtOH/CD formulations, since the stickiness of 0.1% CDs was not felt (approximately 8 × 10^−3^ N). The DEET residue on the skin superficial layers was prolonged, and the drug penetration into the skin tissue was decreased by the addition of 0.1% CD. In particular, the retention time and attenuated penetration of DEET on the rat skin treated with the DEET/EtOH/HPβCD formulation was significantly higher in comparison with that of the DEET/EtOH formulation without CD. Moreover, the repellent effect of DEET was more sustained by the addition of 0.1% HPβCD in the study using *Aedes albopictus*. In conclusion, we found that the DEET/EtOH/HPβCD formulations reduced the skin penetration of DEET and prolonged the repellent action without stickiness.

## 1. Introduction

Along with a mosquito bite comes the discomfort of an allergic reaction with skin rashes, edema, and pruritus, and the risk of disease transmission. Recently, mosquitoes such as *Culex* sp., *Anopheles* sp., *Aedes albopictus*, and *Aedes aegypti* have expanded their range due to the global warming: they can travel between nations, have adapted to the urban environment, and increased the incidence of diseases caused by mosquito bites around the world. In order to prevent insects landing and biting, insect repellent formulations and window screens in buildings and open environments are useful as effective preventive measures. The most widely used synthetic repellents are *N*,*N*-diethyl-3-toluamide (DEET), *N*,*N*-diethylphenylacetamide (DEPA), permethrin (synthetic pyrethroid), ethyl utylacetylaminopropionate (IR3535, EB), 1-(1-ethylpropoxycarbonyl)-2-(2-hydroxyethyl) (picaridin or icaridin), neem, citronella, and lemon eucalyptus essential oil [1]. In particular, DEET is the most efficient and is the one of repellents referred to by the World Health Organization [2].

DEET is a slightly yellowish, oily, volatile liquid, with a faint odor at room temperature. Moreover, DEET is very soluble in alcohol, moderately soluble in petroleum ether, and insoluble in water. It is hypothesized that the mechanism of repellent action by DEET is related to affecting the chemoreceptors that inhibit the biting behavior after a mosquito’s contact with the skin and the binding of the DEET to the olfactory receptors of mosquitoes, thus repelling them from skin while still in flight [3,4]. Briefly, first hypothesis states that DEET odor activates specific olfactory receptor neurons (ORNs) activating various glomeruli in antennal lobe and it is processed in lateral horn (LH) leading to aversive behavior. Human sweat is perceived by specific olfactory receptors thereby activating ORNs which provide signals to the antennal lobe. The processing of signals occurs in the lateral horn (LH) leading to attraction towards the host. The second model explains that DEET acts as an inhibitor of attractive odorants from host when it perceives odor by blocking the activation of ORNs which cannot be processed further in the LH. The third hypothesis states that modulation of the OR complex occurs where DEET odor modulates the perception in glomeruli in the antennal lobe, thereby making confusion in odor perception. Fourth, stimulation of gustatory receptor neurons (GRNs) occurs by tasting DEET by labellum which leads to the activation of bitter GRNs. Fifth, Inhibition of sweet GRNs occurs on tasting DEET with labellum [3,4,5]. The DEET is incorporated into topical formulation from 7–10% (short periods, approximately 2 h) to 20–30% (longer periods, approximately 6 h) [6]. With proper application, the use of DEET is allowed in concentrations of up to 10% for children of 2–12 years, and the safety record of DEET has proven to be excellent [7]. However, it can be absorbed by the skin and pass through the cutaneous barrier, reaching deeper layers and blood vessels, since DEET is a lipophilic feature [8]. Therefore, the presence of DEET in the bloodstream was observed with prolonged exposure and high concentrations and should be avoided by pregnant women, because excessive DEET in blood causes adverse effects, such as central nervous system toxicity, encephalopathy, seizures, and skin rash [9,10,11]. In addition, excessive DEET has also been shown to damage painted surfaces, plastics, leather, and synthetic fabrics [7,11]. Thus, insect repellents have topical repellent action, and the minimum permeation of DEET is favored as well as the retention of the repellent activity in the superficial layers of the skin [12].

Several different formulations have been studied to improve the efficacy and safety of these repellents, such as nanoemulsions, liposomes, solid lipid micro- and nanoparticles, microcapsules, and polymer blends [13,14,15,16,17]. Moreover, the latest study has shown that its efficacy also increases by the synergism between very low doses with other compounds, such as IR3535 [18]. The complex with cyclodextrins (CDs) is another technique used to modify the release of volatile repellent activity. The CDs are cyclic carbohydrates with a hydrophilic outer surface and a hydrophobic central cavity that are widely used as complexes [19,20]. Three types of CDs based on 6, 7, and 8 D-glucopyranose units are classified as α-CD, β-CD, and γ-CD, respectively. Among these CDs, the β-CD is applied to many fields, such as catalysts, complexes [21], pharmaceuticals [22], and separation [23], although the application of β-CD is limited by its low solubilization. In order to overcome the shortcomings of the β-CD, chemical modification of β-CDs with various functional groups, such as 2-hydroxypropyl-β-CD (HPβCD), methyl-β-CD (MeβCD), and sulfobutylether-β-CD (SBEβCD) have been extensively used. On the other hand, a previous study reported that CD-grafted cotton fabric showed a controlled release of DEET [24]; however, the system using CD-grafted cotton fabric provided a weak repellent activity against mosquitoes. This behavior was related to the slow DEET evaporation from the matrix, likely as a result of a strong interaction between the repellent and the reservoir side (skin and fabric) [24]. Moreover, it was reported that using water containing 6–30% CDs (CD/water) as a solvent instead of ethanol (EtOH) enhanced the repellent action time duration of DEET formulation [25,26,27,28]; although, the CD/water with high-dose CD caused high stickiness [24], and the stickiness limited its use. In this study, we tried to find a CD concentration that would not be sticky and attempted to prepare a DEET formulation by the combination of EtOH and low-dose CD as solvents (DEET/EtOH/CD formulations). In addition, we investigated the characteristics, drug behavior on the skin, and repellent action in DEET/EtOH/CD formulations.

## 2. Results

### 2.1. Changes in Characteristics of DEET/EtOH Formulations by the Addition of CDs

Stickiness is a major factor in the feel of insect repellents. In Figure 1, we investigated the relationships between CD and tensile stress (stickiness) by using mice skin. Previous reports showed that the use of 6–30% CDs solution as a solvent of DEET instead of EtOH led to the enhancement of the repellent action time duration [25,26,27,28]. Therefore, we measured the tensile stress of 0.1–30% HPβCD solution. The tensile stress was enhanced with an increase in the amount of HPβCD, and the 30% HPβCD solution provided a high tensile stress of 18.0 × 10^−3^ N. However, the addition of the 0.1% HPβCD solution also tended to increase the tensile stress in comparison with the vehicle, the tensile stress was lower, since the tensile stress levels in 0.1% HPβCD solution were 0.4-fold of that in the 30% HPβCD solution. Next, we measured the changes in tensile stress of 0.1% of β-CD, HPβCD, MeβCD, and SBEβCD solutions. The tensile stress of 0.1% MeβCD was higher than that of the corresponding HPβCD. In contrast to the result of the MeβCD solution, the tensile stress of the 0.1% β-CD and SEBβCD solutions were similar to that of the corresponding HPβCD solution. On the contrary, the tensile stress of the DEET/EtOH/CD formulations with 0.1% β-CD, HPβCD, and SEBβCD (Rp. 2, 3, and 5 formulations) showed no significant difference with the DEET/EtOH (Rp. 1 formulation). The tensile stress of the DEET/EtOH/HPβCD formulation containing 30% HPβCD was 22 × 10^−3^ N (*n* = 5). Furthermore, we demonstrated the viscosity in the DEET/EtOH/CD formulations (Figure 2). The viscosity of 0.1% HPβCD solution was lower than that of 30% HPβCD solution, and the viscosities in the DEET/EtOH formulations containing 0.1% CDs were similar to formulations without 0.1% CDs, and no significant difference in viscosity was observed in the DEET/EtOH formulations with or without 0.1% CDs. Figure 3 shows the effect of CD on DEET adsorption in a polypropylene test tube. The DEET (Rp. 1 formulation) was adsorbed on the polypropylene test tube, and the adsorbed amount was 4.43% 3 weeks after preparation at 4 °C. On the other hand, the addition of CD attenuated the adsorption of DEET on polypropylene, and the adsorbed amount of DEET in the Rp. 2–Rp. 5 formulations were 0.87%, 0.01%, 3.34%, and 1.77%, respectively, at 4 °C. The adsorption of DEET tended to decrease at 25 °C; however, no significant difference in drug adsorption was observed in the DEET/EtOH formulations with or without CDs at 25 °C and 50 °C. On the other hand, the degradability of DEET was not observed in the DEET/EtOH formulations with or without 0.1% CD for 3 weeks in this study.

### 2.2. Changes in the Skin Penetration and Repellent Action of DEET/EtOH Formulations by the Addition of CDs

The DEET retention on the skin and its repellent action in the DEET/EtOH/CD formulations were also investigated in this study. Figure 4 shows the changes in volatile volume of DEET/EtOH/CD formulations. The addition of CDs reduced the volatile volume (Figure 4A), and a significant difference in the volatile volume was observed between DEET/EtOH formulations with or without CDs (Figure 4B). On the contrary, the retention of DEET in the superficial layers of the skin (skin surface and stratum corneum) were improved by the addition of HPβCD, and the DEET value (*AUC*_0–4h_) in the Rp. 3 formulation with HPβCD was 3.2-fold of the Rp. 1 formulation without CD (Figure 5). In addition, the penetration into the skin tissue (epithelium and dermis) was decreased by the addition of β-CD, HPβCD, and SBEβCD (Figure 6). In particular, it was confirmed that the addition of HPβCD improved the retention at the superficial layers of the skin and decreased the penetration into the tissue. Based on these results of drug retention in the superficial layers of the skin (Figure 5 and Figure 6), we investigated the changes in the repellent action of DEET in DEET/EtOH formulations containing 0.1% HPβCD, which is considered to be highly practical, by using *Aedes albopictus* (Figure 7). The repellent rate in the DEET/EtOH formulations containing HPβCD was less than 90% 5 h after application. On the other hand, the repellent effect of DEET was sustained by the addition of HPβCD, and the repellent rate in DEET/EtOH/HPβCD dropped below 90% 7 h after application. Thus, the addition of 0.1% HPβCD prolonged the repellent effect of DEET.

## 3. Discussion

With a mosquito bite comes the risk of disease transmission; hence, a safe and effective topical repellent formulation is required for the prevention of biting activity. DEET is one of the most widely used insect repellents in the world, and we aimed to improve the feel of the product, extend the duration of action, and enhance its safety. In this study, we attempted to design a DEET formulation using the combination of EtOH and 0.1% CD as solvents (DEET/EtOH/CD formulation). In addition, we found that the stickiness of this formulation was minimized, the DEET/EtOH/CD formulations reduced drug transdermal penetration, and they prolonged the repellent effect in comparison with the DEET/EtOH formulation without CD.

The repellent formulation should be active on the superficial layer of skin and provide slow evaporation, resulting in long-lasting protection. β-CD (MW 1134.9, melting point 209–300 °C) and the chemical modification of β-CD (HPβCD, MeβCD, and SBEβCD) have been extensively used in various fields as catalysts, complexes, pharmaceuticals, and separation [21,22,23]. HPβCD (MW 1541.5, melting point 278 °C), the most widely used modified CD, has excellent inclusion properties for many compounds, is an effective drug carrier, and is safe and less toxic [29,30,31,32,33]; MeβCD (MW 1303.3, melting point 180–182 °C) is water soluble, biodegradable, cheap, weakly toxic [34], could be widely applied in chemical reactions as an inverse phase transfer catalyst [35], and could be applied in the enhancement of bioavailability [36], stabilization [37], and solubilization. Moreover, SBEβCD (MW 1451, melting point 202–204 °C) has been widely used as a chemical modification of β-CDs. Previous studies using β-CD and HPβCD reported that 6–30% CDs can form a complex with the repellents decreasing evaporation and, hence, enhancing the repellent action time duration [25,26,27,28]. However, the solution containing 6–30% CDs caused high stickiness [24], and the stickiness limited its use. Therefore, we tried to find a CD concentration that would not be sticky. The stickiness (tensile stress) of MeβCD was higher than that of HPβCD. On the other hand, the stickiness was decreased with the reduction in HPβCD content, and the stickiness of the vehicle and the 0.1% HPβCD solution were about the same (Figure 1). In addition, the stickiness of DEET/EtOH formulations containing 0.1% CDs (β-CD, HPβCD, and SBEβCD) were also similar to the DEET/EtOH formulation without CD, and no significant difference in viscosity was observed in the DEET/EtOH formulations with or without 0.1% CDs (Figure 2). From these findings, we determined the CD concentration to be 0.1% in the DEET/EtOH/CD formulations. On the other hand, the adsorption of DEET on containers influences its storage methods. Therefore, we investigated the effect of 0.1% CDs on DEET adsorption in a polypropylene test tube. The DEET was adsorbed on a polypropylene test tube at 4 °C, 25 °C, and 50 °C. However, no significant difference in drug adsorption was observed in the DEET/EtOH formulations with or without CDs at 25 °C and 50 °C, the adsorption of DEET on polypropylene was prevented by the combination of EtOH and 0.1% β-CD and HPβCD at 4 °C (Figure 3). This result showed that the solvent consisting of EtOH and 0.1% β-CD and HPβCD makes the storage of DEET easier under low temperatures. However, we do not have detailed data on why the addition of 0.1% β-CD and HPβCD suppressed DEET adsorption at 4 °C condition. It is possible that the lower thermal energy at 4 °C results in less adsorption, and that the suppression effect of CD may be more expressed. Further study is needed on these mechanisms.

It is important to attenuate the penetration of the skin to avoid entry into the blood in the development of an insect repellent [38]. β-CD is composed of seven-(1,4)-linked glucopyranose units, arranged surrounding a slightly lipophilic cavity in a peculiar truncated-cone-shaped structure, and the CDs can accommodate guest molecules through the formation of inclusion complexes [39]. In addition, the formation of inclusion complexes by CD prolonged the retention of DEET on the skin and resulted in the controlled-release system [24]. In this study, the volatile volume of formulation based on DEET/EtOH/CD was decreased in comparison with the DEET/EtOH without CD (Figure 4). The evaporation rate of the repellent vehicle is an important factor for the time of action against the insects, and the reduction in the evaporation rate may cause the prolongation of the repellent activity of DEET. Moreover, the permeation trials showed that the DEET content in the superficial layers of the skin treated with DEET/EtOH/HPβCD was higher than that in DEET/EtOH without CD, since the DEET residue on the skin (skin surface and stratum corneum) in the Rp. 3 formulation was significantly higher and the penetration into the skin tissue (epithelium and dermis) was lower than that in the Rp. 1 formulation (Figure 5 and Figure 6). These results showed that the solvent consisting of EtOH and 0.1% HPβCD was able to reduce blood bioavailability, making the product developed more secure. It was reported that the combination of polyglycerol (PG) and polyethylene glycol 400 (PEG 400) with DEET had shown reduced permeation in skin as well by altering the skin/vehicle partition coefficient [40]. It is possible that the reduction in skin permeation of DEET may be also related these changes of skin/vehicle partition coefficient. On the other hand, the HPβCD was more effective in the retention of DEET in the superficial layers of the skin and the inhibition of penetration into the skin tissue than other CDs. A previous study reported that the transdermal absorption of the drug was enhanced by a MW of less than 500, a melting point of less than 200 °C, and an oil–water partition coefficient of 1–4 [41,42,43,44]. The MW and melting point of HPβCD is 1541.5 and 278 °C, respectively, and the values are higher than that of other CDs. Moreover, the solubility of HPβCD is also higher in comparison with other CDs [21,22,23,29,30,31,32,33,34,35,36,37]. These characteristics of HPβCD might provide benefits as a vehicle for DEET, such as increased skin retention and drastic reduction in repellent skin permeation.

Furthermore, we investigated whether the formulation based on DEET/EtOH/HPβCD increased repellent activity. It is important to select the appropriate model animals for a study evaluating the repellent activity of DEET. The *Aedes albopictus* is type of mosquito that has high activity in the daytime and prefers to suck human blood. The *Culex pipiens* and *Culex p. molestus* FORSKAL are highly active only at dusk and dawn. Therefore, the *Aedes albopictus* with a long activity time was selected as the model animal. It is known that for practical use, the repellent effect of DEET should be 80% or more (90%). In this study, the repellent effect of DEET was sustained by the addition of HPβCD, and the repellent rate of the DEET/EtOH/HPβCD formulation remained at 80% 7 h after application (Figure 7). Thus, the addition of 0.1% HPβCD prolonged the repellent effect of DEET. Katz et al. [38] reported that not penetrating the skin, avoiding entry into the bloodstream, and a 7–8 h effect are the ideal characteristics of an insect repellent. These DEET formulations using EtOH and low-dose HPβCD as vehicles may be useful in developing ideal insect repellent formulations.

Further studies are needed to investigate how the skin permeation of DEET is attenuated by the addition of low-dose CDs. Moreover, it is important to clarify whether there is any synergistic repellent effect of DEET with EtOH and CD. In a future work, we plan to measure the synergistic repellent effect of DEET/EtOH/HPβCD formulation. In addition, recent studies provide the evidence that insect-specific ORs and GRs are possible direct targets. Therefore, future work is also needed to demonstrate the molecular targets of DEET.

## 4. Materials and Methods

### 4.1. Chemicals

DEET was provided from Combi-Blockes Inc. (San Diego, CA, USA). β-CD, MeβCD and EtOH was purchased from Wako Pure Chemical Industries, Ltd. (Osaka, Japan). HPβCD was obtained from Nihon Shokuhin Kako Co., Ltd. (Tokyo, Japan). SBEβCD was provided from Funakoshi Co., Ltd. (Tokyo, Japan). All other chemicals were of the highest purity commercially available.

### 4.2. Animals

Seven-week-old male ICR mice (approximately 37 g) and six-week-old male Wistar rats (approximately 200 g) were purchased from the Kiwa Laboratory Animals Co., Ltd. (Wakayama, Japan), and female *Aedes albopictus* were provided by Nagasaki University and raised by Earth Corporation. The mice and rats were housed at 25 °C under normal conditions and provided with water and a standard diet CE-2 (Clea Japan Inc., Tokyo, Japan) freely. All experiments for animals were performed according to the guidelines of Kindai University and the Japanese Pharmacological Society. The experiments were approved on 1 April 2019 by Kindai University, and the project code is KAPS-31-001 and KAPS-31-010.

### 4.3. Preparation of DEET/EtOH/CD Formulations

Ten percent DEET was mixed with 40% EtOH containing 0.1% CDs. The four CDs (β-CD, HPβCD, MeβCD, and SBEβCD) were selected, and the compositions of DEET/EtOH/CD formulations prepared are shown in Table 1. The EtOH has been used commonly solvent, since purified water and DEET is not mix. From these results, we used DEET/EtOH formulation as a standard formulation (control) in this study.

### 4.4. Characterization of DEET/EtOH/CD Formulations

The viscosity at 25 °C was measured by an SV-1A (A&D Company, Limited, Tokyo, Japan). The DEET concentrations were determined using an LC-20AT HPLC system (Shimadzu Corp., Kyoto, Japan) with a 2.1 × 50 mm Inertsil^®^ ODS-3 column at 35 °C (GL Science Co., Inc., Tokyo, Japan). Five microgram per milliliter butyl p-hydroxybenzoate in methanol was selected as an internal standard, and for the wavelength for detection, a mobile phase was used of 210 nm and 30% acetonitrile (*v*/*v*%), respectively. The mobile phase flowed at 0.3 mL/min. The DEET and butyl p-hydroxybenzoate were detected at 6.25 min and 13.5 min, respectively. Commercially available insect repellents are stored under different living conditions. Therefore, we evaluated the adsorption of DEET between 4–50 °C, a common living environment temperature in this study. In the measurement of the drug adsorption on the tube, 5 mL of each DEET/EtOH/CD formulation was added to the 15 mL polypropylene test tube (drug adsorption), and stored at 4 °C, 25 °C, and 50 °C for 3 weeks. After that, the concentration of DEET was measured by using the HPLC method described above. The amount of drug adsorption on the polypropylene (polypropylene test tube) was calculated from the difference in the concentration of DEET before and after the storage in this study [DEET content (%) = DEET level_3 weeks_/DEET level_0 week_ × 100].

### 4.5. Tensile Stress of DEET/EtOH/CD Formulations

Tensile strength was measured according to the instructions of a Force Tester MCT-2150 (A & D Co., LTD., Osaka, Japan). The dorsal and abdominal hair of seven-week-old male ICR mice was removed 1 day before their use in experiments, the mice were euthanized by injecting a lethal dose of pentobarbital, and the skin was carefully separated from other ocular tissues. The removed skins were trimmed to 25 cm^2^ (5 cm × 5 cm), and the two pieces of skin were overlapped. The DEET/EtOH/CD formulations (1.67 µL/cm^2^) were treated between the overlapped skins, and the tensile stress (stickiness) was calculated by pulling with a Force Tester MCT-2150 at 25 °C.

### 4.6. Water Retention in DEET/EtOH/CD Formulations

Fifty microliters of DEET/EtOH/CD formulations were smeared on absorbent cotton, and incubated at 25 °C for 20 min. After that, the weight of the absorbent cotton treated with DEET/EtOH/CD formulations was measured. In this study, the water volatility (g) was calculated from the difference between the pre- and post-incubation, and the residual volume was expressed as a ratio to the pre-incubation amount.

### 4.7. Measurement of DEET on the Skin Surface, Stratum Corneum, Epithelium, and Dermis of Rats

The DEET level in skin tissue were evaluated by method partially modified the in vitro drug penetration test using the Franz diffusion cell to mimic living conditions [45,46,47]. The abdominal hair of 7-week-old Wistar rats was removed 1 d before their use in experiments, and the rats were euthanized by injecting a lethal dose of pentobarbital; then, the abdominal skin was carefully separated from other ocular tissues. The removed skin (2.01 cm^2^) was set on a filter paper moistened with saline solution, and 1.67 µL/cm^2^ of DEET/EtOH/CD formulations were treated on the skin surface side. After that, the skins were incubated at 37 °C for 2 h and 4 h. Then, the skin was separated into skin surface, stratum corneum, epithelium, and dermis. In this study, the DEET in the stratum corneum was collected by the tape-stripping method. These tissues were homogenized in 300 µL of methanol and were centrifuged (20,400× *g*, 20 min, 4 °C). The DEET in the supernatants was measured by the HPLC method described above, and the trapezoidal rule was used to calculate the area under the skin concentration–time curve (*AUC*_0–4h_).

### 4.8. Measurement of the Repellent Effect of DEET

One cage (25 × 25 × 25 cm, metal mesh cage) was used for each subject and about 50 female adult mosquitoes (*Aedes albopictus*, transferred from Nagasaki University and raised by Earth Corporation, 7 to 14 days after emergence) were added. After treating with 1.67 µL/cm^2^ of the formulations (DEET/EtOH or DEET/EtOH/HPβCD formulations) to the back of one hand (treated area), we cut out some of the rubber glove by about 5 × 5 cm and wore it. The other hand wore a similar glove (untreated area). We put the untreated and treated area in the cages in order and counted the number of mosquitoes that landed on the exposed skin in order (5 min, 25–30 °C). This operation was repeated three times by multiple subjects, and the repellent rate was calculated by the following formula.
(1)$$\mathrm{Repellent}\;\mathrm{rate}\;\left(\%\right)=\left(1-\frac{\mathrm{number}\;\mathrm{of}\;\mathrm{landings}\;\mathrm{in}\;\mathrm{treated}\;\mathrm{area}\;}{\mathrm{number}\;\mathrm{of}\;\mathrm{landings}\;\mathrm{in}\;\mathrm{untreated}\;\mathrm{area}}\right)\times 100$$

### 4.9. Statistical Analysis

The data are expressed as mean ± standard error (S.E.). A statistical analysis was performed using the Student’s *t*-test and ANOVA followed by Dunnett’s multiple comparison, with *p* < 0.05 considered to be significant.

## 5. Conclusions

We designed a DEET formulation using low-dose CD (0.1%) and EtOH as solvents (DEET/EtOH/CD formulations) and found that the DEET/EtOH/HPβCD became safer due to reduced skin permeation with a prolonged repellent action. The authors concluded that the combination of EtOH and low HPβCD is a promising solvent for DEET due to the increased skin retention and drastic reduction in repellent skin permeation, and the DEET/EtOH/HPβCD is useful as a release system for repellent formulations (Figure 8). Thus, this kind of DEET/EtOH/HPβCD may eventually be helpful for minimizing health problems.

## Figures and Tables

**Figure 1 molecules-27-03174-f001:**
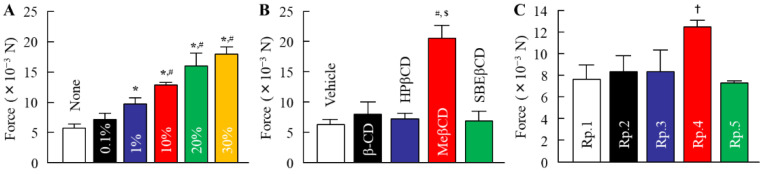
Effect of each CD on the tensile stress (stickiness) in the DEET/EtOH/CD formulations. (**A**) Changes in tensile stress of the 0.1–30% HPβCD solution. (**B**) Changes in tensile stress of the various 0.1% CD solutions. (**C**) Changes in tensile stress of the DEET/EtOH/CD (10:40:0.1%) formulations (Rp. 1–5 formulations). The compositions of DEET/EtOH/CD formulations (Rp. 1–5 formulations) are shown in Table 1. *n* = 5–8. * *p* < 0.05 vs. None for each category. ^#^
*p* < 0.05 vs. 0.1% HPβCD for each category. ^$^
*p* < 0.05 vs. Vehicle for each category. ^✝^
*p* < 0.05 vs. Rp. 1 formulation for each category. Although, the tensile stress was enhanced with HPβCD solution contents, no significant tensile stress was observed in the DEET/EtOH formulations with or without 0.1% HPβCD.

**Figure 2 molecules-27-03174-f002:**
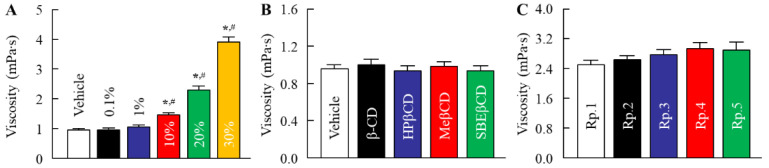
Effect of each CD on the viscosity in the DEET/EtOH/CD formulations. (**A**) Changes in viscosity of the 0.1–30% HPβCD solution. (**B**) Changes in viscosity of the various 0.1% CD solutions. (**C**) Changes in viscosity in the DEET/EtOH/CD formulations (Rp. 1–5 formulations). The compositions of DEET/EtOH/CD (10:40:0.1,%) formulations (Rp. 1–5 formulations) are shown in Table 1. *n* = 5. * *p* < 0.05 vs. Vehicle for each category. ^#^
*p* < 0.05 vs. Rp. 1 formulation. The viscosity in the DEET/EtOH formulations containing 0.1% CD (Rp. 2–5 formulations) were not observed to be significantly different in comparison with formulations without CD (Rp. 1 formulation).

**Figure 3 molecules-27-03174-f003:**
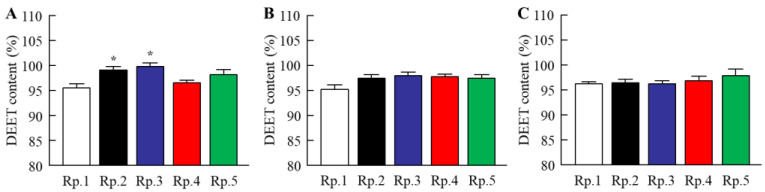
Changes in drug adsorption on polypropylene in the DEET/EtOH/CD (10:40:0.1%) formulations at 4 °C (**A**), 25 °C (**B**), and 50 °C (**C**). The compositions of DEET/EtOH/CD formulations (Rp. 1–5 formulations) are shown in Table 1, and the DEET/EtOH formulations were stored for 3 weeks. *n* = 8. * *p* < 0.05 vs. Rp. 1 formulation for each category. The adsorption of DEET was significantly prevented by the addition of 0.1% β-CD and HPβCD at the 4 °C condition. On the other hand, no significant difference in drug adsorption was observed in the DEET/EtOH formulations with or without CDs at 25 °C and 50 °C.

**Figure 4 molecules-27-03174-f004:**
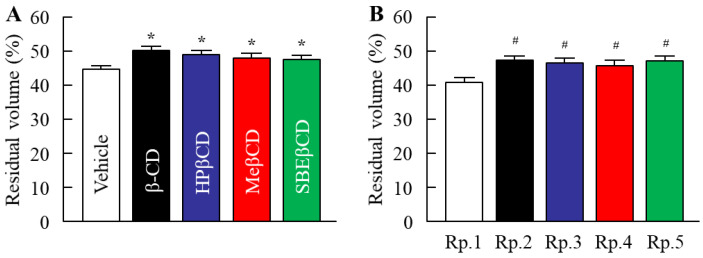
Effect of each CD on the volatile volume of the DEET/EtOH/CD formulations. (**A**) Changes in volatile volume of the various 0.1% CDs solutions. (**B**) Changes in volatile volume in the DEET/EtOH/CD (10:40:0.1,%) formulations (Rp. 1–5 formulations). The compositions of DEET/EtOH/CD formulations (Rp. 1–5 formulations) are shown in Table 1. *n* = 8. * *p* < 0.05 vs. Vehicle. ^#^
*p* < 0.05 vs. Rp. 1 formulation. The volatile volume of DEET/EtOH formulations was decreased by the addition of 0.1% CDs.

**Figure 5 molecules-27-03174-f005:**
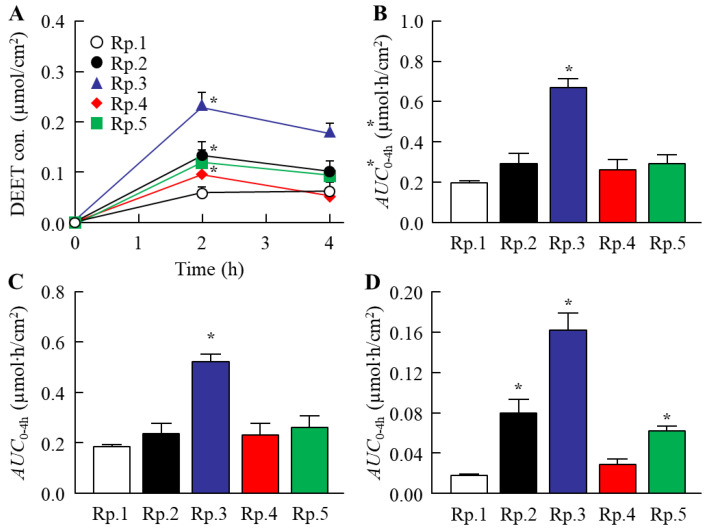
Effect of each CD on the retention of DEET in the superficial layers of the skin treated with DEET/EtOH/CD (10:40:0.1,%) formulations. (**A**,**B**) Changes in DEET contents (**A**) and *AUC*_0–4h_ (**B**) in the superficial layers of the skin (skin surface and stratum corneum). (**C**,**D**) Changes in *AUC*_0–4h_ of the skin surface (**C**) and stratum corneum (**D**). The compositions of DEET/EtOH/CD formulations (Rp. 1–5 formulations) are shown in Table 1. *n* = 5–7. * *p* < 0.05 vs. Rp. 1 formulation for each category. The addition of β-CD, HPβCD, and SBEβCD enhanced the contents of DEET in the stratum corneum of the skin. In particular, the HPβCD prolonged the retention of DEET in the superficial layers.

**Figure 6 molecules-27-03174-f006:**
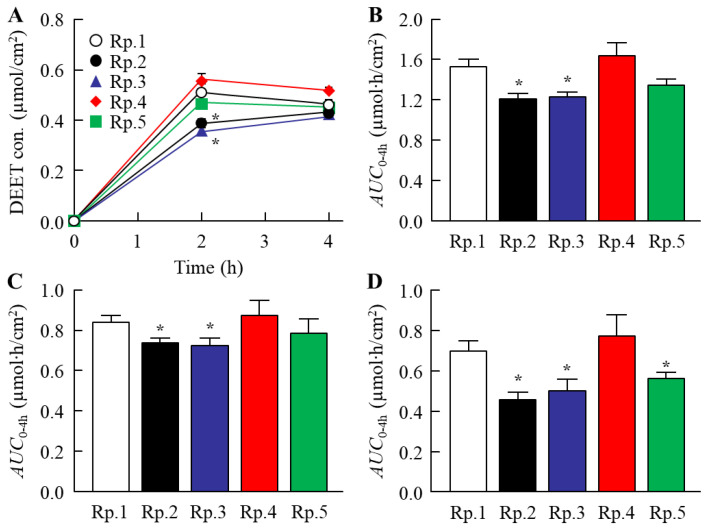
Effect of each CD on DEET penetration in the skin of rats treated with DEET/EtOH/CD (10:40:0.1%) formulations. (**A**,**B**) Changes in DEET contents (**A**) and *AUC*_0–4h_ (**B**) of skin consisting of epithelium and dermis. (C) and (D) Changes in *AUC*_0–4h_ of skin epithelium (**C**) and dermis (**D**). The compositions of DEET/EtOH/CD formulations (Rp. 1–5 formulations) are shown in Table 1. *n* = 5–7. * *p* < 0.05 vs. Rp. 1 formulation for each category. The addition of β-CD, HPβCD, and SEBβCD decreased the DEET penetration into the skin tissue.

**Figure 7 molecules-27-03174-f007:**
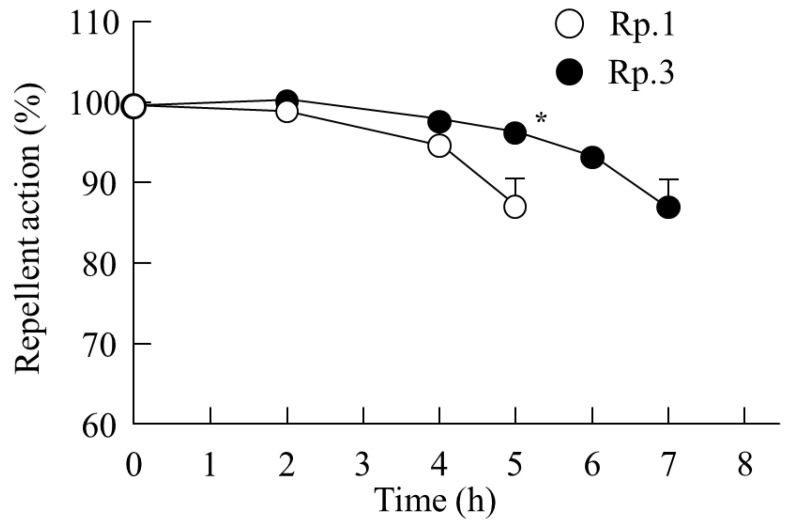
Effect of 0.1% HPβCD on the repellent action of DEET/EtOH (10:40%). The compositions of Rp. 1 and Rp. 3 formulations are shown in Table 1. *n* = 6. * *p* < 0.05 vs. Rp. 1 formulation. The addition of 0.1% HPβCD prolonged the repellent effect of DEET/EtOH.

**Figure 8 molecules-27-03174-f008:**
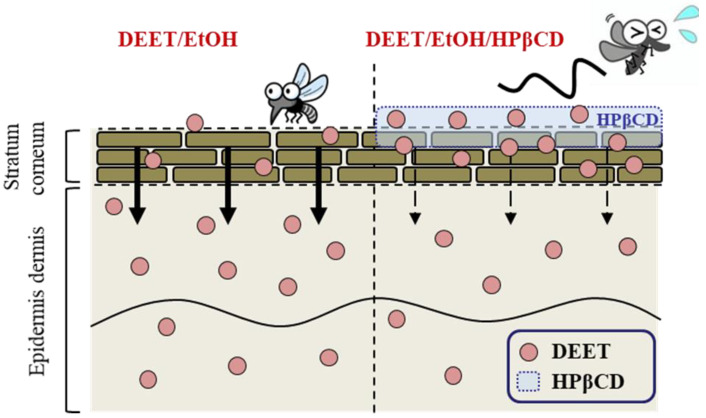
Graphical image for repellent action after the treatment with an insect repellent formulation based on DEET/EtOH/HPβCD.

**Table 1 molecules-27-03174-t001:** Formulations of DEET/EtOH/CD in this study.

Formulation	Content (*w*/*v*%)						
	DEET	EtOH	β-CD	HPβCD	MeβCD	SBEβCD	Distilled Water ad.
Rp. 1 DEET/EtOH	10	40					100
Rp. 2 DEET/EtOH/βCD	10	40	0.1				100
Rp. 3 DEET/EtOH/HPβCD	10	40		0.1			100
Rp. 4 DEET/EtOH/MeβCD	10	40			0.1		100
Rp. 5 DEET/EtOH/SBEβCD	10	40				0.1	100

Rp. 2–5 were shown as DEET/EtOH/CD formulation.

## Data Availability

Data is contained within the article.

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
