# Peer review of "N,N-Diethyl-3-toluamide Formulation Based on Ethanol Containing 0.1% 2-Hydroxypropyl-β-cyclodextrin Attenuates the Drug’s Skin Penetration and Prolongs the Repellent Effect without Stickiness"

_molecules, 2022, doi:10.3390/molecules27103174_

Round 1

Reviewer 1 Report

Dear Authors,

You have done interesting research but some improvemsnts could be done.

First, in titel and abstract DEET abrevation should not be used. 

Througout complete text ethanol is suggested not to be benefitial however only formulation with ethan is investigated.

Why not research without ethanol was condacted.

Figure 8 does not represent mechanisam but it could be graphival absttact.

In methods origin of wistar rats is not described. In characterisation was not mentionated how many repetation of experiments wqs done. 

Also why different temperatures were used in drug adsorption. Deet degradability should be mentioned in results.

Tensil strength and penetration is done according to what previous described methods or didt you develop it (it should be determined).

It shod be also stated time intervals for penetration.

Best of luck

Author Response

We carefully revised our manuscript according to the suggestions of the reviewer 1, and details are as follows.

<Q and A for Reviewer 1>

Q1. First, in title and abstract DEET abrevation should not be used

A1. The reviewer’s comment is correct. In order to respond to the reviewer’s comment, we showed the full spell (N,N-diethyl-3-toluamide) of DEET (title and abstract).

Q2. Througout complete text ethanol is suggested not to be benefitial however only formulation with ethan is investigated. Why not research without ethanol was condacted.

A2. Thank you for pointing out this. DEET is liquid at room temperature, and is diluted with a solvent when the DEET is used. The ethanol have been used commonly solvent, since purified water and DEET is not mix. From these results, we used DEET/ethanol formulation as a control. In order to respond to the reviewer’s comment, we added the contents in the Materials and Methods (line 330-332).

Q3. Figure 8 does not represent mechanisam but it could be graphival absttact.

A3. Thank you very much for pointing this out. In order to respond to the reviewer’s comment, we revised to “Graphical image for repellent action after the treatment with an insect repellent formulation based on DEET/EtOH/HPβCD” (Figure 8 legend title).

Q4. In methods origin of wistar rats is not described. In characterisation was not mentionated how many repetation of experiments wqs done.

A4. The Wistar rats (approximately 200 g) were purchased from the Kiwa Laboratory Animals Co., Ltd. (Wakayama, Japan). Moreover, we showed the number in the Fig. legend. We added the contents in the Materials and Methods (line 317-319, Figure legend).

Q5. Why different temperatures were used in drug adsorption. Deet degradability should be mentioned in results.

A5. Commercial available insect repellents are stored under different living conditions. Therefore, we evaluated the adsorption of DEET between 4ºC-50ºC, a common living environment temperature in this study. In addition, we added the result of DEET degradability in the result. Thank you very much for your great comment (line 345-347).

Q6. Tensil strength and penetration is done according to what previous described methods or didt you develop it (it should be determined).

A6. The reviewer’s comments are very important. Tensile strength was measured according to the instructions of a Force Tester MCT-2150 (A & D Co., LTD., Osaka, Japan). Moreover, the DEET level in skin tissue were evaluated by method partially modified in vitro drug penetration test using the Franz diffusion cell to mimic living conditions. In order to respond to the reviewer’s comment, we added the information in the Material and Methods (line 358-359, 376-377, Reference 45-47).

Q7. It shod be also stated time intervals for penetration.

A7. The penetration were measured 2 h and 4 h after the application of DEET formulation. In order to respond to the reviewer’s comment, we mentioned the time intervals for penetration in the Material and Methods (line 383).

Thank you for great comments.

Reviewer 2 Report

Nagai et al. study the DEET formulation based on the solvent to enhance the repellent action. The author shows that the use of low dose cyclodextrin (CD) as a solvent increases DEET efficacy as a repellent with longer durability time. Further, to reduce the stickiness due to addition of CD, they mixed the ethanol-based CD with DEET. The application with above formulation on the rat’s skin had attenuation of penetration with enhancement of retention time and have lower stickiness. This signifies that addition of low dose of CD in ethanol based DEET mixture is very useful to reduce the penetration of DEET from skin thereby prolonging the repellency effect. Overall, the study is well designed with sequential study of stickiness, viscosity, adsorption, and effect of ethanol-based CD with DEET formulation.  

Overall, the results in the manuscript are concise and are interesting to the scientific community. However, a couple of issues should be addressed before publication.

Major concerns

  1. Introduction should include the mode of DEET action. There are multiple hypotheses to explain the mode of DEET action. First hypothesis states that DEET odor activates specific ORNs activating various glomeruli in antennal lobe and it is processed in lateral horn (LH) leading to aversive behavior. Human sweat is perceived by specific olfactory receptors thereby activating olfactory receptor neurons (ORNs) which provide signals to the antennal lobe. The processing of signal occurs in the lateral horn (LH) leading to attraction towards the host. The second model explains that DEET acts as an inhibitor of attractive odorants from host when it perceives odor by blocking the activation of ORNs which cannot be processed further in the LH. The third hypothesis states that modulation of OR complex occurs where DEET odor modulates the perception in glomeruli in antennal lobe thereby making confusion in odor perception. Forth, Stimulation of gustatory receptor neurons (GRNs) occurs by tasting DEET by labellum which leads to the activation of bitter GRNs. Fifth, Inhibition of sweet GRNs occurs on tasting DEET with labellum. ------Please see recent review for the mode of DEET action (I don’t know that your first citation is relevant, I can’t not find this).

Cellular and molecular mechanisms of DEET toxicity and disease-carrying insect vectors: a review. Shrestha B, Lee Y. Genes Genomics. 2020 Oct;42(10):1131-1144

  1. What is the molecular target of DEET? Recent studies provide the evidence that insect-specific ORs and GRs are possible direct targets. I wonder if there is any synergistic repellent effect of DEET with EtOH and CD. However, it is not a target study, but at least recommend you to explain any possible effect in Discussion.
  2. You should present figures with recent style with dot-plot, so you allow readers to see each data point.
  3. Line 62-65 describes the different formulation and complex of DEET to improve its efficacy and safety level. Latest study has shown that its efficacy also increases by the synergism between very low dose with other compounds like IR3535 (Cellular and molecular basis of IR3535 perception in Drosophila. Pest Manag Sci. 2022 Feb;78(2):793-802). Therefore, include recent background further.
  4. This study only compares the ethanol-based formulation. Previously combination of PEG 400 and PG with DEET had shown reduced permeation in skin as well (Ross et al 2000). What do you think the durability of these combination with 0.1% CD in compared with ethanol-based formulation? Have you ever tested the use of CD with some other solvents? If possible, I recommend you to increase N number. 3 trials are too low. Increase at least 5 trials for all the case.
  5. Figure 3, is there any reason to check the adsorption after 3 weeks? Have you checked it at different days or week? What happens before and after 3 weeks? There should be any rationale to point out one specific time point.

Minor concerns

  1. Line 34-35, expand the reason for increment of disease incidence as urbanization is not only sole reasons. For, e.g. global warming is also another cause.
  2. Is there any approximate time for reaching DEET particle in blood vessels after direct exposure in skin?
  3. I suggest including the concentration of DEET and ethanol in mixture in the result section as well for readers although it is stated in the method.
  4. Regarding figure label, I would like to suggest making label in similar position for all figures. For e.g. A to align at first major point of Y-axis.
  5. Line 137, ETOH>EtOH.
  6. Line 224, polyethylene>polypropylene.
  7. Line 228-229, “….makes the storage of DEET easier under low temperature”. Further explanation for the justification of this statement is required.

Author Response

We carefully revised our manuscript according to the suggestions of the reviewer 2, and details are as follows.

<Q and A for Reviewer 2>

Major concerns

Q1. Introduction should include the mode of DEET action.

A1. The reviewer’s comment is correct. We added the mode of DEET action according to reviewer 2 comments and reference in the Introduction (line 51-63, Reference 5).

Q2. What is the molecular target of DEET? Recent studies provide the evidence that insect-specific ORs and GRs are possible direct targets. I wonder if there is any synergistic repellent effect of DEET with EtOH and CD. However, it is not a target study, but at least recommend you to explain any possible effect in Discussion.

A2. The reviewer’s comments are very important. In order to respond to the reviewer’s comment, we added the importance to investigate about the molecular target of DEET and any possible synergistic repellent effect of DEET with EtOH and CD in Discussion. Thank you for pointing out this (line 300-307).

Q3. You should present figures with recent style with dot-plot, so you allow readers to see each data point.

A3. The reviewer’s comment is correct. The dot-plot style have used for the clinical research and study with many number (n). On the other hand, the basic research using experimental animal is presented as mean ± S.E. (the many Fig in other MDPI journal and high journals, such as nature, science and cell, are also presented as mean ± SE in the basic research). This may change in the future for basic research, but for this presentation we would like to use the mean ± S.E., which is generally used in our field of research. Thank you very much for pointing this out.

Q4. Line 62-65 describes the different formulation and complex of DEET to improve its efficacy and safety level. Latest study has shown that its efficacy also increases by the synergism between very low dose with other compounds like IR3535 (Cellular and molecular basis of IR3535 perception in Drosophila. Pest Manag Sci. 2022 Feb;78(2):793-802). Therefore, include recent background further.

A4. Thank you for pointing out this. In order to respond to the reviewer’s comment, we added the latest study and reference introduced by the reviewer in the Introduction (line 79-81, Reference 18).

Q5. This study only compares the ethanol-based formulation. Previously combination of PEG 400 and PG with DEET had shown reduced permeation in skin as well (Ross et al 2000). What do you think the durability of these combination with 0.1% CD in compared with ethanol-based formulation?

A5. The durability of these combination with 0.1% CD tend to enhance, but no significant difference observed between DEET formulation with or without CDs (ethanol-based formulation) at 4ºC-50ºC. In addition, the degradability of DEET was not observed in the DEET/EtOH formulations with or without 0.1% CD for 3 weeks in this study. In order to respond to the reviewer’s comment, we added these results and cited the result about previously combination of PEG 400 and PG with DEET (line 133-135, 271-275, Reference 40).

Q6. Have you ever tested the use of CD with some other solvents?

A6. We have experimental data on the addition of CD and l-menthol to DEET/ethanol, but we do not have results on the addition of CD to solvents other than ethanol. The addition of l-menthol to DEET/ethanol/CD has been shown to improve the stickiness of the DEET formulation. These results will be reported in the next paper. Thank you very much for your great comments.

Q7. If possible, I recommend you to increase N number. 3 trials are too low. Increase at least 5 trials for all the case.

A7. The reviewer’s comments are very important. In order to respond to the reviewer’s comment, we increased N number to over 5 trials. It is known that for practical use, the repellent effect of DEET should be 80% or more (90%). Therefore, we consider the data in Rp.1 to be fine up to 5 h of application in the Fig. 7. From these findings, we removed the data of Rp.1 at 6 h and 7 h after application in the Fig. 7 (Figure).

Q8. Figure 3, is there any reason to check the adsorption after 3 weeks? Have you checked it at different days or week? What happens before and after 3 weeks? There should be any rationale to point out one specific time point.

A8. Thank you for pointing out this. The results at the beginning of the experiment (3 weeks ago) are shown as 100% in the Fig. 3. The DEET content was measured over time (1 d, 3 d, 5 d, 1 week, 2 weeks, and 3 weeks), and the results after 3 weeks were shown in this study, since the most adsorption was observed. In order to respond to the reviewer’s comment, we added the information in the Material and Methods (line 354-355).

Minor concerns

Q9. Line 34-35, expand the reason for increment of disease incidence as urbanization is not only sole reasons. For, e.g. global warming is also another cause.

A9. In order to respond to the reviewer’s comment, we added the more information about the reason for increment of disease incidence in the Introduction (line 36).

Q10. Is there any approximate time for reaching DEET particle in blood vessels after direct exposure in skin?

A10. The rate of drug absorption into the bloodstream varies depending on the solvent used, but most transdermal formulation are detected in the body within 3 h after transdermal application. Thank you very much for your great comments.

Q11. I suggest including the concentration of DEET and ethanol in mixture in the result section as well for readers although it is stated in the method.

A11. The reviewer’s comments are very important. We added the concentration of DEET and ethanol in mixture in the Fig. legend (Figure legend).

Q12. Regarding figure label, I would like to suggest making label in similar position for all figures. For e.g. A to align at first major point of Y-axis.

A12. The reviewer’s comment is correct. In order to respond to the reviewer’s comment, we revised Figure label according to reviewer comment. Thank you for pointing out this (Figure).

Q13. Line 137, ETOH>EtOH. Line 224, polyethylene>polypropylene.

A13. Thank you for pointing out this. We collected these spell miss.

Q14. Line 228-229, “….makes the storage of DEET easier under low temperature”. Further explanation for the justification of this statement is required.

A14. The reviewer’s comment is correct. However, we do not have detailed data on why the addition of CD suppressed DEET adsorption under low temperature conditions. It is possible that the lower thermal energy at 4℃ results in less adsorption, and that the suppression effect of CD is more expressed. Further research is needed to investigate these issues. In order to respond to the reviewer’s comment, we added these contents in the Discussion (line 250-253).

Thank you for great comments.

Round 2

Reviewer 2 Report

I still recommend using the dot-plot and mean with SEM together. This is not only used in the field of clinical science, but also most journals ask it. If authors do not have the Graph Prism, I alternatively ask authors to provide all the raw data in supplementary table. Therefore, readers can see each data point. 

Author Response

We carefully revised our manuscript according to the suggestions of the reviewer 2, and details are as follows.

<Q and A for Reviewer 2>

Q1. I still recommend using the dot-plot and mean with SEM together. This is not only used in the field of clinical science, but also most journals ask it. If authors do not have the Graph Prism, I alternatively ask authors to provide all the raw data in supplementary table.

A1. In order to respond to the reviewer’s comment, we showed all the raw data. Thank you very much for your comment.

Thank you for great comments.
